# The Factors That Affect Turnover Intention According to Clinical Experience: A Focus on Organizational Justice and Nursing Core Competency

**DOI:** 10.3390/ijerph19063515

**Published:** 2022-03-16

**Authors:** Hanna Choi, Sujin Shin

**Affiliations:** College of Nursing, Ewha Womans University, Seoul 03760, Korea; hnchoi1105@ewhain.net

**Keywords:** organizational justice, clinical competency, personnel turnover

## Abstract

The purpose of this study was to investigate organizational justice and nursing core competency as factors that affect turnover intention among nurses. The participants comprised 153 nurses who worked at tertiary hospitals and general hospitals throughout South Korea. Turnover intention was measured using the Nurse Turnover Intention Scale. Organizational justice was measured using the Justice Scale, and nursing core competency was measured using the Korean Nursing Core Competency Scale. Data were collected via an online survey and analyzed using multiple regression. Among nurses with less than 3 years of clinical experience, a lower distributive justice score (β = −0.47, *p* < 0.001) was associated with high turnover intention. Among nurses with 3 to 6 years of clinical experience, a lower interactional justice score (β = −0.37, *p* = 0.042) and high nursing core competency (β = 0.31, *p* = 0.034) were associated with high turnover intention. The type of organizational justice that influenced turnover intention differed depending on clinical experience. These results highlight the need to understand the professional characteristics of nurses according to their clinical experience and to provide targeted organizational support and effective competency-based human resource management.

## 1. Introduction

The demand for high-quality medical care and skilled nurses is gradually increasing due to social and demographic changes [1]. However, the number of nurses per 1000 people, which is an indicator of nurse staffing levels, is 4.2 in South Korea, which is only just over half of the average value (7.9) of Organization for Economic Cooperation and Development countries [2]. Furthermore, in South Korea, the employee turnover rate of nurses is 15.2%, which is more than seven times higher than the average for other healthcare professionals [3], and 40% of clinical nurses in South Korea have less than 3 years of clinical experience [4]. In contrast, the average age of nurses in the United States is 40 to 50 years old, with 30% of clinical nurses having almost 10 years of clinical experience [5]. A high employee turnover rate reduces organizational capacity [6], thereby affecting the quality of nursing care and ultimately posing a threat to patient safety [7]. This is why addressing the employee turnover rate is an important component of managing a hospital’s workforce.

Turnover intention, one of the major predisposing factors for a job change [8], has been used as a variable in multiple studies on job changes [9,10,11]. Turnover intention refers to the intention of a member of an organization to leave his or her current job or position [12]. Organizational factors such as organizational justice, organizational commitment, and workplace violence, and personal factors such as resilience, self-efficacy, and occupational stress were reported as factors influencing nurses’ turnover intention [13]. Until now, studies have mainly focused on the effect of personal factors on turnover intention, but recent studies have shown that organizational factors are also important in explaining nurses’ turnover intention [14], and organizational justice is one of the most important organizational factors influencing nurses’ turnover intention among organizational factors [15,16,17].

Organizational justice refers to the degree to which one perceives his or her compensation, organizational decision-making processes, and interactions within the organization to be fair [18]. The three types of organizational justice are distributive justice, procedural justice, and interactional justice. Distributive justice refers to one’s perception of fairness related to compensation according to his or her level of education, effort, and competency. Procedural justice refers to one’s perception of fairness concerning the decision-making processes of an organization. Lastly, interactional justice refers to one’s perception of fairness concerning interactions within an organization [19]. Previous studies on organizational justice have reported that more clinical experience corresponded to a lower degree of perceived organizational justice [20]. This suggests that nurses often believe that they are not compensated adequately according to their clinical competency or responsibilities. It has also been reported that nurses considered changing jobs due to stress they experienced related to administrative decisions and communication within the workplace [21]. The studies, as a whole, highlight the importance of organizational justice, which examines the fairness of compensation, decision-making processes, and interactions within the organization. Individuals who consider their organizations to be unfair tend to have high turnover intention [9,22] and little motivation to be proactive about the growth of their organization [23].

Nonetheless, even when an organization treats its nurses fairly, the turnover intention of individual nurses varies, which is why nursing core competency is an influential situational factor. Nursing core competency is an integral element for executing high-quality, professional nursing care and includes factors such as technical skills, attitudes, motivation, personal insight, interpretation ability, receptivity, and self-assessment [24]. Although the term “nursing core competency” is sometimes used interchangeably with work performance, nursing competency, and clinical competency, it has recently been redefined to refer to factors that are prerequisites for providing professional nursing care. Identifying and developing nursing core competency is a critical component of operating an organization [25] since it directly relates to building a qualified nursing workforce and ensuring high-quality nursing care. However, since the nursing core competency varies among nurses, individual perceptions regarding the fairness of compensation, decision-making processes, and interactions within an organization, and, consequently, the individual turnover intention of nurses, differ in relation to their competency. In particular, nurses with high nursing core competency struggle with workload, work stress, and exhaustion associated with training recently graduated nurses and other responsibilities held by highly competent nurses [26]. Fair procedures and interactions, as well as compensation that reflects individual competency, are essential, but the implementation of these elements tends to be insufficient in actual workplaces [27]. Only a few studies have investigated the correlations of nursing core competency with organizational justice and turnover intention, even though turnover intention in relation to nursing core competency is vital for building a skilled nursing workforce.

Therefore, this study aimed to examine the effects of perceived organizational justice and nursing core competency on the turnover intention of clinical nurses in order to provide baseline data for effective management of the nursing workforce.

## 2. Materials and Methods

### 2.1. Study Design

This is a descriptive study examining the effects of nurses’ perceived organizational justice and nursing core competency on turnover intention.

### 2.2. Sample

The participants in this study were nurses who actively worked at Korean medical institutions. Specifically, the participants were nurses who worked at tertiary hospitals or general hospitals nationwide. For the purpose of this study, nurses who were on a leave of absence were excluded. In addition, since the criteria for nursing core competency in this study emphasized the roles and responsibilities of general ward nurses, outpatient nurses and surgical nurses were excluded. The number of study participants was computed using G*Power 3.1.9.7 (IBM Corp., Armonk, NY, USA). Based on a previous study by Kim and Kim [13], the minimum sample size required for a multiple regression analysis using a significance level of 0.05, effect size of 0.20, power of 0.95, and 15 independent variables was 153 participants. Considering a possible 10% dropout rate due to missing or incomplete data, 170 participants were recruited. Seventeen participants whose responses were insufficient were excluded, and 153 participants were selected as the final subjects for the data analysis.

### 2.3. Measures

#### 2.3.1. General Characteristics

Information on personal characteristics including gender, age, marital status, education level, clinical experience, current position, type of work, and department, as well as organizational characteristics including the hospital type and bed-to-nurse ratio, were collected through a survey.

#### 2.3.2. Organizational Justice

The Organizational Justice Scale, which was developed by Niehoff and Moorman [18] and translated into Korean by Jung and Choi [28], was used after permission was obtained from the original authors and translators via email. The scale includes 5 questions on distributive justice, 6 questions on procedural justice, and 9 questions on interactional justice, comprising a total of 20 questions. The score for each question was calculated using a 5-point Likert scale in which a score of 1 indicated “strongly disagree” while a score of 5 indicated “strongly agree” with a minimum possible score of 20 points and a maximum possible score of 100 points. A high score indicated a high level of perceived organizational justice. Cronbach’s α in Niehoff and Moorman’s [18] study was greater than 0.90. Cronbach’s α in Jung and Choi’s [28] study was 0.96, and it was 0.94 in this study.

#### 2.3.3. Nursing Core Competency

The Korean Nurse’s Core Competency Scale (KNCCS) developed by Lee et al. [29] was used after obtaining permission and receiving the Korean version of the scale from the authors via email. The KNCCS includes 21 questions on understanding of human beings and communication, 13 questions on professional attitudes, 14 questions on critical thinking and evaluation, 13 questions on general clinical performance, and 9 questions on special clinical performance, comprising a total of 70 questions. The subcategory on general clinical performance includes questions on the performance of basic nursing responsibilities such as drug administration and providing basic nursing care, whereas the subcategory on special clinical performance refers to the performance of emergency nursing, intensive care, and end-of-life care. Each question is answered using a 5-point Likert scale, with 1 point indicating “not able to perform” and 5 points indicating “able to perform with confidence” with a minimum possible score of 70 and a maximum possible score of 350. A high score indicates a high degree of nursing core competency. The reliability score, as indicated by Cronbach’s α, was 0.97 at the time of the tool’s initial development by Lee et al. [29] and 0.98 in this study.

#### 2.3.4. Turnover Intention

The Nurse Turnover Intention Scale developed by Yeun and Kim [30] was used after obtaining permission from the authors via email. The scale includes 4 questions on job satisfaction, 3 questions on work performance, and 3 questions on interpersonal relationships, comprising a total of 10 questions. The contents of the question are whether the subject intends to move his or her job when there is difficulty related with each subscale. Each question is answered using a 5-point Likert scale, with 1 point indicating “strongly disagree” and 5 points indicating “strongly agree” with a minimum possible score of 10 and a maximum possible score of 50. A high score indicates a high degree of turnover intention. The reliability score as indicated by Cronbach’s α was 0.83 at the time of the tool’s initial development by Yeun and Kim [30] and 0.83 in this study.

### 2.4. Data Collection

Data were collected through an online survey to ensure voluntary participation and the participants’ anonymity. The terms of the study for participants were included in the survey, including explanations that the identities of participants would not be revealed as a result of participation in the study, that participation may be withdrawn at any time, and that the collected data would be used solely for the purposes of this study. A participant recruitment notice was published from 21 October 2021 to 28 October 2021, in the form of a digital banner on a community accessed through a web portal where nurses exchange information. Upon clicking the recruitment notice, participants could voluntarily choose to participate in the online survey.

### 2.5. Data Analysis

The collected data were analyzed using SPSS 28.0 (IBM Corp., Armonk, NY, USA). The participants’ general characteristics, perceived organizational justice scores, nursing core competency scores, and turnover intention scores were analyzed in terms of frequency, percentage, average, and standard deviation. To analyze the differences in organizational justice, nursing core competency, and turnover intention scores in relation to the participants’ general characteristics, the *t*-test and one-way analysis of variance were used, and the post hoc Scheffé test was performed. The correlations between key research variables were analyzed using Pearson correlation coefficients, and the effects of organizational justice and nursing core competency on turnover intention were analyzed via multiple regression analysis.

### 2.6. Ethical Considerations

This study was conducted with approval of the institutional review board (**** 2021-08-031-004) of the authors’ institution. Participants were required to read and agree to the terms. Only those who agreed and provided informed consent were asked to participate in the survey, and the survey automatically ended if the participant did not agree to the terms and provide informed consent.

## 3. Results

### 3.1. General Characteristics and Their Effects on Turnover Intention

Among the 153 participants, 92.2% (*N* = 141) were women and 64.1% (*N* = 98) were between the ages of 25 and 30. A total of 86.3% (*N* = 132) of participants were not married, and the highest education level for 83.7% (*N* = 128) of the participants was a bachelor’s degree. In total, 37.2% (*N* = 57) of participants had 3 to 6 years of clinical experience, and 84.3% (*N* = 129) of participants were clinical nurses. A total of 83.7% (*N* = 128) of participants worked third-shift positions. A total of 54.9% (*N* = 84) of participants worked in tertiary hospitals, and the average bed-to-nurse ratios were 1.98:1 for general wards, 0.70:1 for emergency rooms, and 0.50:1 for intensive care units. No differences in turnover intention according to the participants’ general characteristics were observed. However, the further Scheffé test on ‘type of work’ shows that the individual means of each group do not differ (Table 1).

### 3.2. Participants’ Perceived Organizational Justice, Nursing Core Competency, and Turnover Intention

The average score for perceived organizational justice was 62.25 ± 14.09 out of 100. The scores for the three types of perceived organizational justice (interactional justice, procedural justice, and distributive justice) were 28.84 ± 7.58, 19.28 ± 4.89, and 14.13 ± 3.83, respectively. The average score for nursing core competency was 275.86 ± 34.52, and the scores for the five subcategories of nursing core competency (understanding of human beings and communication, critical thinking and evaluation, general clinical performance, professional attitudes, and special clinical performance) were 84.41 ± 10.26, 55.09 ± 7.22, 52.63 ± 7.08, 49.69 ± 7.87, and 34.03 ± 5.75, respectively. The average score for turnover intention was 41.13 ± 6.40 out of 50, and the scores of the three subcategories of turnover intention (job satisfaction, work performance, and interpersonal relationships) were 15.99 ± 3.24, 12.78 ± 2.06, and 12.36 ± 2.37, respectively (Table 2).

### 3.3. Correlations between Perceived Organizational Justice, Nursing Core Competency, and Turnover Intention

The turnover intention of participants showed a statistically significant negative correlation with perceived organizational justice (r = −0.23, *p* = 0.004). This suggests that low perceived organizational justice is associated with a high degree of turnover intention. In particular, turnover intention showed a statistically significant negative correlation with distributive justice (r = −0.35, *p* < 0.001), which is one of the subcategories of organizational justice (Table 3).

### 3.4. The Effect of Perceived Organizational Justice and Nursing Core Competency on Turnover Intention by Clinical Experience

The analysis results for the effect of perceived organizational justice and nursing core competency in relation to the participant’s clinical experience by organizational justice type are shown in Table 4. Among participants with less than 3 years of clinical experience, distributive justice was negatively correlated (β = −0.47, *p* < 0.001) with turnover intention. Perceived organizational justice accounted for 19% of variance in turnover intention, and the regression model was statistically significant (F = 2.59, *p* = 0.021). In participants with 3 to 6 years of clinical experience, interactional justice showed a negative correlation (β = −0.37, *p* = 0.042) with turnover intention, and nursing core competency exhibited a positive correlation (β = 0.31, *p* = 0.034) with turnover intention. Perceived organizational justice and nursing core competency accounted for 20% of variance in turnover intention, and the regression model was statistically significant (F = 2.71, *p* = 0.015). In participants with more than 6 years of clinical experience, the regression model was not statistically significant (F = 1.90, *p* = 0.087) regarding the effect of perceived organizational justice and nursing core competency on turnover intention by organizational justice type (Table 4).

## 4. Discussion

This study was conducted to collect data to improve the management of the nursing workforce by analyzing the factors that affected turnover intention given the consistently high turnover rate of nurses in South Korea. In this study, distributive justice, which is one of the three subcategories of organizational justice, was identified as a factor that affected the turnover intention of nurses. This finding is similar to that of a previous study suggesting that nurses who perceived their organization’s distributive justice to be low experienced a high degree of turnover intention, indicating that they felt that they were not adequately compensated for their work [14,20]. In order to provide high-quality nursing care to patients, medical service providers must maintain a sufficient staff of healthcare professionals [31]. Although there is no international standard regarding the optimal bed-to-nurse ratio, an appropriate ratio is critical for patient safety, as lower staffing levels are associated with lower patient mortality [32] and lower job exhaustion and job dissatisfaction, as reported by previous studies [33].

Even though nurses are one of the categories of healthcare professionals who provide actual medical services to patients, they tend to receive a relatively low wage compared to other healthcare professionals [3]. This correlates to the findings of previous studies that confirmed that the perceived organizational justice of nurses tended to be lower compared to that of other healthcare professionals, particularly doctors and pharmacists [34]. These findings suggest that nurses view their own compensation to be low given their working hours and the intensity of the work. This study confirmed that distributive justice had a negative correlation with turnover intention, and it is vitally important to devise and implement institutional measures to ensure an appropriate intensity of work, bed-to-nurse ratio, and level of compensation for nurses.

In this study, the subcategories of organizational justice that affected turnover intention differed according to participants’ clinical experience. Since the characteristics of nurses vary depending on their length of clinical experience [35], a differentiated analysis by clinical experience is required. For nurses with less than 3 years of clinical experience, distributive justice was the most influential factor affecting turnover intention. This was similar to the results of another study that found that lower salary satisfaction correlated to a higher degree of turnover intention among new graduate nurses [36]. Nurses with less than 3 years of clinical experience tend to be recent graduates of nursing school [35], and they are usually assigned a heavy workload that includes general technical nursing responsibilities. The results of the present study show that distributive justice influenced turnover intention, and, given the characteristics of nurses with less than 3 years of clinical experience, the findings suggest a need for the appropriate distribution of responsibilities and compensation. Furthermore, nursing core competency was significantly lower among nurses with less than 3 years of clinical experience, highlighting the importance of training and adaptation for nurses who are recent graduates. In addition, it was recently found that workplace violence had a negative effect as a factor influencing work function [36]. As new nurses with less than 3 years are vulnerable to violence in the workplace, various analyses may be needed for the nursing core compliance of new nurses with less than 3 years [37].

This study shows that, among nurses with 3 to 6 years of clinical career, perceived interactional justice and nursing core competency were the factors that most strongly influenced turnover intention. This finding is similar to that of a previous study in which lower perceived interactional justice was associated with a higher degree of turnover intention [9]. Nurses with 3 to 6 years of clinical experience may be considered “proficient” nurses [35] and are likely to take on administrative roles such as that of a preceptor [38]. Nevertheless, their limited involvement in the decision-making process or lack of promotions despite increased responsibilities can lead to low perceived interactional justice and a corresponding high degree of turnover intention. Moreover, among nurses with 3 to 6 years of clinical experience, a high degree of nursing core competency was associated with a high degree of turnover intention. This finding is different from that of a previous study in which high nursing core competency was found to correspond to a low degree of turnover intention [39], although a direct comparison is difficult due to the small number of comparable studies. The findings of this study may indicate that there is a lack of competency assessment and appropriate compensation for nurses with 3 to 6 years of clinical experience, thus leading to a high turnover rate [28].

This finding may be credited to the fact that most Korean hospitals have an annual salary system or a step-based salary system [27] that does not necessarily reflect one’s individual performance and degree of competency. In Korea, competency-based assessments for nurses are either insufficient or are limited to a formality. In other countries, however, competency-based human resources management systems have been found to enhance nursing competency as well as job satisfaction and retention intention [25]. As was the case in some Korean hospitals that adopted an experience ladder system to implement a compensation plan that reflects individual competency, competency-based human resources management systems increase nursing core competency [40]. This type of system may be an effective way to manage talent within an organization.

In most of the studies, nursing core competency is used to evaluate the ability of nursing students [40,41]. However, since developing competency is crucial for nurses, who are frontline healthcare workers and manage the health of patients, studies that assess nursing core competency of nurses are more needed. Although nurses with 3 to 6 years of clinical experience play important roles as trainers and competent staff members at hospitals, there is little evidence-based research regarding the appropriate level of compensation and positions for nurses of this experience level; therefore, more studies should examine nursing competency.

The distribution of clinical experience in this study was similar to that of a status report by the Korea Hospital Nurses Association [4] that found that 40% of nurses had less than 3 years of clinical experience, 13.4% had between 3 to 5 years of clinical experience, and 47.4% had more than 5 years of clinical experience. Since nurses with less than 3 years of clinical experience are considered recent nursing school graduates according to the 5-level classification system of nurses’ clinical proficiency [35], and 30% to 40% of nurses currently in the workforce have less than 3 years of clinical experience, the percentage of nurses who are recent graduates in this study appears to be considerably high. An even distribution of nurses with different degrees of career proficiency is essential for the growth of an organization. Therefore, institutional strategies such as career ladder or competency-based human resource management should be prepared to balance the proportion of nurses who are recent graduates and experienced nurses.

Of the participants’ general characteristics, current position seems to have played a role in participants’ perceived organizational justice. This is similar in part to the findings of a previous study [42] in which perceived procedural justice was found to be high among nurses with higher positions such as head nurses. This may be because higher-ranking nurses have the authority to make decisions and have fewer superiors. This highlights the importance of building an organizational culture that can be perceived as being equally just from the perspective of general nurses.

The bed-to-nurse ratio, as an organizational characteristic, was used in this study to measure the staffing level of nurses. This parameter was used because it is potentially difficult to calculate the number of patients for whom each nurse is responsible in nursing units that operate according to a functional nursing system. The average bed-to-nurse ratio in this study was 1.98:1 for general wards, 0.70:1 for emergency rooms, and 0.50:1 for intensive care units. Using the system of nurse staffing grades for tertiary hospitals, given that the majority of the study participants worked at tertiary hospitals, general wards on average were considered to have grade 1 staffing, emergency rooms were considered to have grade 3 staffing, and intensive care units were considered to have grade 2 staffing in this study.

Currently, South Korea has no laws regulating the bed-to-nurse ratio, and the system of nurse staffing grades has been adopted by some hospitals as a way to manage the nursing workforce. The criterion for grade 1 is the presence of fewer than two beds per nurse. This number, however, does not reflect the professional reality for nurses, who work in shifts. Considering that at least five nurses are needed to assign one nurse per shift for a three-shift work schedule, a grade 1 medical institution, in reality, may have only one nurse to provide care for almost 10 patients during any given shift. In addition, this standard is observed in only 37.6% of all hospitals nationwide [4]. As such, the system of nurse staffing grades is not sufficient for maintaining a desirable bed-to-nurse ratio. In the United States, however, legally mandatory maximum bed-to-nurse ratios have been established. The state of California limits the number of patients per nurse to fewer than five per internal medicine unit [43], which is almost half the number of patients according to grade 1 of South Korea’s system.

The system of nurse staffing grades is the most commonly implemented method for managing the nursing workforce of Korean medical institutions, and hospitals seeking to maximize their profits use this system to calculate their revenue relative to the cost of employing their nursing staff. More than half of hospitals do not hire enough nurses [4], but there is no legislation to regulate medical institutions with inadequate nursing personnel. Employing an adequate number of competent nurses is a fundamental component of providing high-quality nursing care and a crucial factor for ensuring patient wellness [32], which is why Korean employment and labor regulations for nurses must be reexamined. Efforts should be made to legally mandate a maximum bed-to-nurse ratio in order to regulate the number of patients per nurse and improve the quality of patient care.

## 5. Limitations

There are some limitations to the study. As survey research, although we have followed the empirically appropriate survey procedure, self-selection could not be perfectly resolved. This study only examines nurses working at general hospitals or territory hospitals nationwide, so there is a limit to generalizing the results of this study to all nurses. In addition, the turnover intention measurement tool was developed in 2013, and there is a limitation in reflecting the current situation, including the COVID-19 situation.

## 6. Conclusions

As a descriptive study designed to investigate the effect of nurses’ perceived organizational justice and nursing core competency on turnover intention, this study demonstrated differences in the effects of the subcategories of organizational justice on turnover intention according to clinical experience. This study has practical significance by offering fundamental data to support the development of strategies for nurse workforce management. Since the participants of this study were nurses who worked in general hospitals or tertiary hospitals nationwide, the findings of this study cannot be generalized to all hospital nurses. Therefore, a study targeting nurses who work at small- to medium-sized hospitals is needed. In addition, due to the small number of studies on nursing core competency targeting experienced nurses, follow-up studies to assess nursing core competency among experienced nurses are recommended.

In this study, high nursing core competency was correlated with a high degree of turnover intention among nurses with 3 to 6 years of clinical experience. This is likely due to a lack of proper assessment and compensation that reflects the competency levels of individual nurses. This study thereby suggests that a competency assessment and compensation system based on clinical experience should be considered to reduce turnover intention among skilled nurses.

Low perceived distributive justice was a factor that influenced turnover intention. Hiring an adequate number of nurses is important for ensuring quality nursing care. For this reason, this study suggests that the current nursing workforce management system must be reconsidered, and legislation should be implemented to regulate the bed-to-nurse ratio of hospitals.

## Figures and Tables

**Table 1 ijerph-19-03515-t001:** General characteristics and turnover intention (*N* = 153).

Variables	Categories	*N* (%)	M ± SD	Turnover Intention		
				M ± SD	t or F	*p*
**Personal characteristics**						
Gender	Female	141 (92.2)		41.23 ± 6.47	0.64	0.524
	Male	12 (7.8)		40.00 ± 5.89		
Age (years)	<25	21 (13.7)	28.58 ± 5.67	40.60 ± 7.09	2.20	0.090
	25 ≤ 30	98 (64.1)		41.94 ± 5.80		
	30 ≤ 35	13 (8.5)		40.77 ± 9.43		
	≥35	21 (13.7)		38.10 ± 5.80		
Marital status	Single	132 (86.3)		41.36 ± 6.49	1.10	0.275
	Married	21 (13.7)		39.71 ± 5.88		
Education level	3-year diploma	6 (3.9)		41.33 ± 7.69	2.47	0.088
	Bachelor’s degree	128 (83.7)		41.58 ± 6.17		
	Master’s degree or above	19 (12.4)		38.11 ± 7.20		
Clinical experience (years)	<3	54 (35.3)	5.50 ± 5.80	41.07 ± 6.41	0.81	0.445
	3–5.99	57 (37.2)		41.88 ± 5.62		
	≥6	42 (27.5)		40.21 ± 7.41		
Position	Staff nurse	129 (84.3)		41.67 ± 6.38	1.35	0.180
	Charge nurse or above	24 (15.7)		39.47 ± 5.79		
Type of work	3-shift	128 (83.7)		41.80 ± 6.11	3.47	0.018
	Day/evening	15 (9.8)		37.80 ± 7.12		
	Non-shift	8 (5.2)		36.38 ± 7.58		
	Other	2 (1.3)		42.50 ± 0.71		
**Organizational characteristics**						
Hospital type	General hospital	69 (45.1)		41.64 ± 6.08	0.87	0.384
	Tertiary hospital	84 (54.9)		40.73 ± 6.69		
Bed-to-nurse ratio	General ward		1.98:1			
	Emergency room		0.70:1			
	Intensive care unit		0.50:1			

**Table 2 ijerph-19-03515-t002:** Organizational justice, nursing core competency, and turnover intention (*N* = 153).

Variables	M ± SD	Range	Min	Max
**Organizational justice (overall)**	62.25 ± 14.09	20–100	25	92
Interactional justice	28.84 ± 7.58	9–45	9	45
Procedural justice	19.28 ± 4.89	6–30	7	30
Distributive justice	14.13 ± 3.83	5–25	5	23
**Nursing core competency**	275.86 ± 34.52	70–350	188	350
Understanding of human beings and communication	84.41 ± 10.26	21–105	58	105
Critical thinking and evaluations	55.09 ± 7.22	14–70	37	70
General clinical performance	52.63 ± 7.08	13–65	34	65
Professional attitudes	49.69 ± 7.87	13–65	29	65
Special clinical performance	34.03 ± 5.75	9–45	21	45
**Turnover intention**	41.13 ± 6.40	10–50	20	50
Job satisfaction	15.99 ± 3.24	4–20	8	20
Work performance	12.78 ± 2.06	3–15	6	15
Interpersonal relationships	12.36 ± 2.37	3–15	6	15

M, mean; Max, maximum; Min, minimum; SD, standard deviation.

**Table 3 ijerph-19-03515-t003:** Correlations between turnover intention, organizational justice, and nursing core competency (*N* = 153).

Variables	Turnover Intention	
	r	*p*
Organizational justice	−0.23	0.004
Interactional justice	−0.15	0.070
Procedural justice	−0.16	0.053
Distributive justice	−0.35	<0.001
Nursing core competency	0.03	0.770

**Table 4 ijerph-19-03515-t004:** Factors influencing turnover intention by clinical experience (*N* = 153).

Variables	<3 Years (*n* = 54)				3–5.99 Years (*n* = 57)				≥6 Years (*n* = 42)			
	B	β	t	*p*	B	β	t	*p*	B	β	t	*p*
(constant)	69.99		4.34	<0.001	42.77		2.78	0.008	41.41		3.61	0.001
Distributive justice	−0.95	−0.47	−3.45	0.001	−0.38	−0.27	−1.80	0.079	−0.98	−0.56	−2.26	0.031
Procedural justice	−0.19	−0.14	−0.65	0.517	−0.05	0.03	0.17	0.867	0.12	0.09	0.36	0.722
Interactional justice	0.18	0.19	0.88	0.384	−0.29	−0.37	−2.09	0.042	0.21	0.25	0.90	0.373
Nursing core competency	0.00	−0.00	−0.01	0.993	0.06	0.31	0.03	0.034	0.04	0.19	0.94	0.356
Age	−0.68	−0.16	−1.21	0.231	−0.13	−0.03	0.81	0.806	−0.10	−0.09	−0.41	0.681
Education (diploma) ^†^	4.96	0.11	0.82	0.418	−2.74	0.11	0.39	0.390	5.21	0.15	0.88	0.386
Education (≥master) ^†^	11.73	0.25	1.96	0.056	−2.93	−0.13	0.29	0.287	−4.26	−0.27	−1.52	0.139
Type of work (other) ^†^	5.62	0.12	0.92	0.365	−4.79	0.11	0.38	0.381				
Type of work (D/E) ^†^									−4.14	−0.27	−1.20	0.239
Type of work (non-shift) ^†^									−2.02	−0.11	−0.47	0.642
*R* ^2^	0.32				0.31				0.35			
Adj. *R*^2^	0.19				0.20				0.17			
F (*p*)	2.59 (0.021)				2.71 (0.015)				1.90 (0.087)			

Adj. *R*^2^, adjusted *R*^2^; SE, standard error adjusted; D, Day; E, Evening. ^†^ Reference group: Education (bachelor), types of work (3 shifts).

## Data Availability

The data presented in this study are available on request from the corresponding author. The data are not publicly available due to the information contained that could compromise the privacy of research participants.

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
