# Peer review of "The Factors That Affect Turnover Intention According to Clinical Experience: A Focus on Organizational Justice and Nursing Core Competency"

_ijerph, 2022, doi:10.3390/ijerph19063515_

Round 1

Reviewer 1 Report

This was a well-designed study, using validated and appropriate questionnaire instruments. Sample size of 153 was large enough to be meaningful. The writing was clear and well organized, with information presented clearly and findings interpreted appropriately.

Reviewer 2 Report

The authors analyze important topic about about the factors which affect turnover intentions of nurses. However, the manuscript should be revised and corrected.:

1) the authors should be more accurate with concepts and terms which they use in the manuscript. For example, they wrote social justice as a keyword but it is not the same as the concept organizational justice which is used in the text. The authors write "A high score indicated a high level of organizational justice" (line 119), but the instrument which is used in the research is used to assess the perceived organizational justice. Please use the right terms for things about which you write;

2) I have doubt regarding instruments used to assess the turnover intentions in the research. The authors write, that this instrument consist of 3 subscales: job satisfaction, work performance and interpersonal relationships. i agree that these 3 are the predictors of turnover intention, but i disagree to believe, that job satisfaction+work performance+interpersonal relationships=turnover intention, as the authors write. This is very important methodological question.

3) The authors write that "No differences in turnover intention according to the participants’ general characteristics were observed" (lines 184-185), but I see that type of work is significant characteristic for turnover intention (see table 1).

4) finally I would like to invite the authors of the manuscript to revise the introduction and Discussion - the quality of these parts will be better if they will be better structured and more new references will be used.

Reviewer 3 Report

This study aims to relate organizational justice and competence with the intention to leave the job.

  1. The Authors should explain in the Introduction why they chose these variables and not others that are commonly associated with intention to quit, for example, violence at work or occupational stress.
  2. The Authors should indicate in which year they conducted the study. it is important to know if it took place during the pandemic because the stress induced by this event is a relevant factor.
  3. The Authors should be aware that participant self-selection can introduce bias.
  4. The sample size is very modest and leaves many doubts about the results achieved. For example, the authors discuss in lines 322-324 the effect of hospital size on perceived justice. It should be noted that the sample was not constructed in such a way as to ensure the representativeness of the two types of hospitals. Furthermore, the calculation of the minimum sample size did not refer to this type of comparison.
  5. The article is completely missing from the section on limitations and this raises concerns that the authors will ignore them.
  6. The main limitation is the cross-sectional character, which does not allow to infer about causality; therefore, the conclusions that the authors draw with great certainty should instead be formulated with many reservations and in hypothetical form.
  7. Although they did not take workplace violence into account, the authors should say in the Discussion that it can potentially affect the working capacity of nurses and, therefore, their turnover intentions. Many papers deal with this issue. For example, Magnavita N, Heponiemi T, Chirico F. Workplace Violence Is Associated With Impaired Work Functioning in Nurses: An Italian Cross-Sectional Study. J Nurs Scholarsh. 2020 May;52(3):281-291. doi: 10.1111/jnu.12549. Please note that it is not mandatory that the authors cite this specific article and they are welcome to seek alternative manuscripts in the literature that are relevant to the manuscript’s content.

Reviewer 4 Report

Strengths: literature review, methodology, results and conclusions

Weaknesses: nothing to report

Improvement proposal

Line 296 – “Most studies on nursing core competence have been used to assess the learning skills of nursing school students” Authors must specify the studies and which specific learning skill.

Line 313 “Therefore, institutional strategies should be implemented to balance the proportion of nurses who are recent graduates and experienced nurses.” Authors should mention what types of strategies should be implemented.

In conclusion

The value of this study for understanding the topic, limitations and future studies should be included.

Round 2

Reviewer 2 Report

Thank you for your efforts to revise the manuscript. Now it seems better for me.

Reviewer 3 Report

the manuscript has been improved